# Effect of Deep Cryogenic Treatment on Microstructure and Properties of Sintered Fe–Co–Cu-Based Diamond Composites

**Siqi Li** [1,2], **Wenhao Dai** [1,2], **Zhe Han** [1,2], **Xinzhe Zhao** [1,2] **and Baochang Liu** [1,2,3,*]

1   College of Construction Engineering, Jilin University, Changchun 130026, China
2   Key Laboratory of Drilling and Exploitation Technology in Complex Conditions of Ministry of Natural Resources, Changchun 130026, China
3   State Key Laboratory of Superhard Materials, Changchun 130021, China
*   Correspondence: liubc@jlu.edu.cn; Tel.: +86-431-8850-2891

**Abstract:** Metal-matrix-impregnated diamond composites are used for fabricating many kinds of diamond tools. In the efforts to satisfy the increasing engineering requirements, researchers have brought more attention to find novel methods of enhancing the performance of impregnated diamond composites. In this study, deep cryogenic treatment was applied to Fe–Co–Cu-based diamond composites to improve their performance. Relative density, hardness, bending strength, and grinding ratio of matrix and diamond composite samples were measured by a series of tests. Meanwhile, the fracture morphologies of all samples after the bending strength test were observed and analyzed by scanning electron microscope. The results showed that the hardness and bending strength of matrix increased slightly after deep cryogenic treatment. The grinding ratio of impregnated diamond composites exhibited a great increase by 32.9% as a result of deep cryogenic treatment. The strengthening mechanism was analyzed in detail. The Fe–Co–Cu-based impregnated composites subjected to deep cryogenic treatment for 1 h exhibited the best overall performance.

**Keywords:** deep cryogenic treatment; impregnated diamond composites; metal matrix; wear resistance

## 1. Introduction

Metal-matrix-impregnated diamond composites are seeing wide use in the fabrication of diamond tools for cutting, grinding, polishing, and drilling hard and brittle materials such as ceramic, rock, and concrete [1–7]. The metal-matrix-impregnated diamond composites prepared by powder metallurgy consist of two parts: diamond grits and metal matrix. The diamond grits embedded in the metal matrix are the working element, and the metal matrix has a decisive effect on whether or not the diamond grits can function fully and effectively [2,8].

At present, pre-alloyed powders have successfully realized industrial application in diamond tools, including diamond cutting segments and diamond drill bits [9–12]. The component uniformity is one of the advantages of pre-alloyed powders, because each alloying particle includes the entire composition, which means the segregation can be avoided [13]. In addition, the pre-alloying of powders can decrease the activation energy, which is conductive to reducing the sintering temperature, which in turn can avoid diamond graphitization [14]. The Fe–Co–Cu matrix, sintered from pre-alloyed Fe–Co–Cu powders, is usually used as the matrix in diamond tool manufacturing [4,10]. It has high strength and adjustable properties suitable for different kinds of hard materials, ensuring that the diamond grits contact the hard materials effectively, maintaining an abrasive cutting surface. Nevertheless, the hard and complex service conditions such as elevated temperature, high impact

stresses, and hydro-abrasion demand that the mechanical properties and wear resistance be further improved [7,15].

Deep cryogenic treatment (DCT) is a supplement of conventional heat treatment used to modify materials properties in order to enhance their performance [16,17]. Recently, some researchers found that deep cryogenic treatment can improve the hardness, compression strength, fatigue resistance, and wear resistance of tool steels [18–23]. The transformation of retained austenite to martensite, the release of residual stress, and the precipitation of ultra-fine carbide particles are the main mechanisms of cryogenic treatment in tool steels [24]. Moreover, some studies have indicated that deep cryogenic treatment can increase the wear resistance and service life of polycrystalline diamond compact (PDC), which is ascribed to the change of stress state [25]. It can be predicted that deep cryogenic treatment is a promising way to improve the properties of metal-matrix-impregnated diamond composites. However, the effect of deep cryogenic treatment on the microstructure, mechanical properties, and wear resistance of metal-matrix-impregnated diamond composites has been rarely studied.

In this study, deep cryogenic treatment was applied to Fe–Co–Cu-based diamond composites. The effects of deep cryogenic treatment on the microstructure, mechanical properties, and wear resistance of Fe–Co–Cu-based diamond composites were investigated and discussed. This work is aimed at seeking the optimal treatment time to meet the requirements of mechanical properties and wear resistance.

## 2. Materials and Methods

### 2.1. Sample Preparation

Two kinds of samples, including matrix samples with and without impregnated diamond grits, were prepared for the property tests. The composition of the Fe–Co–Cu pre-alloy powder (99.9% purity, average particle size of 15 µm, Qinhuangdao, China) used in this work is given in Table 1. The diamond grits (20 vol% concentration, synthetic, 270–380 µm, Zhengzhou, China) were added into the Fe–Co–Cu pre-alloy powder for preparing diamond composite samples through a three-axle mixer for 4 h. The samples (size: $38 \times 8 \times 5$ mm$^3$) were sintered by hot-pressing in graphite molds at 840 °C and a pressure of 20 MPa for 4 min. Sintering temperature, pressure, and time were selected on the basis of previous research and published literature [4,9,10]. The composition, deep cryogenic treatment process, and mechanical properties of samples are given in Table 2.

**Table 1.** The composition of the Fe–Co–Cu pre-alloy powder used in this work.

| Composition | Fe | Co | Cu |
|---|---|---|---|
| Content (wt%) | 54 | 32 | 14 |

**Table 2.** The composition, deep cryogenic treatment (DCT) process, and mechanical properties of samples.

| Samples | Composition | DCT Time (h) | Relative Density (%) | Hardness (HRC) | Bending Strength (MPa) |
|---|---|---|---|---|---|
| S0 | Matrix | - | 97.3 | 38.1 ± 0.6 | 1620 ± 25 |
| S1 | Matrix | 1 | 97.4 | 39.1 ± 0.5 | 1639 ± 23 |
| S2 | Matrix | 2 | 97.3 | 38.8 ± 0.6 | 1720 ± 29 |
| S3 | Matrix | 3 | 97.5 | 39.1 ± 0.3 | 1747 ± 27 |
| SD0 | Matrix + diamond | - | 95.7 | - | 740 ± 35 |
| SD1 | Matrix + diamond | 1 | 96.7 | - | 796 ± 33 |
| SD2 | Matrix + diamond | 2 | 96.2 | - | 752 ± 30 |
| SD3 | Matrix + diamond | 3 | 96.4 | - | 761 ± 50 |

### 2.2. Deep Cryogenic Treatment

A program-controlled cryogenic system (CDW-196, Jinan, China) was used for the deep cryogenic treatment. The schematic diagram is shown in Figure 1. In this device, liquid nitrogen acting as the coolant flows into the distributor to be uniformly dispersed, and then evaporates into nitrogen. Under the action of fan rotation, the gas flow enters the cryogenic tank and exchanges heat with samples. Then, the relatively hot nitrogen gas is emitted into the atmosphere from the exhaust port. Parameters including temperature, cooling rate, and soaking time can be accurately controlled by the computer. In this experiment, the samples were cooled from room temperature (25 °C) down to −190 °C at the cooling rate of 5 °C/min, holding for 1, 2, and 3 h, respectively, and then warmed up to room temperature at a speed of 1 °C/min.

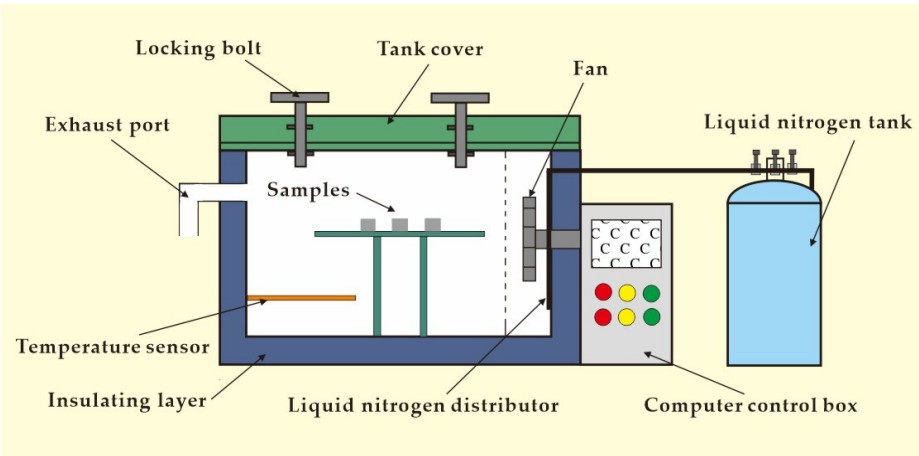

**Figure 1.** The schematic diagram of CDW-196 cryogenic treatment system.

### 2.3. Characterization

The microstructure of composites was characterized by scanning electron microscopy (SEM, Hitachi S-4800, Tokyo, Japan). The phase analysis was evaluated by X-ray diffractometry (TTR-III, Tokyo, Japan). A high-precision density tester (Dahometer, DE-120M, Hangzhou, China) was used to measure the density of samples via the Archimedes method. A Rockwell hardness tester (Huayin HRS-150, Yantai, China) was used to measure the Rockwell hardness scale C (HRC) of samples. Bending strength for each sample was measured by using a three-point bending test machine (DDL 100, Changchun, China), and was calculated by following formula:

$$\sigma = 3PL/2bh^2, \tag{1}$$

where $\sigma$ is the bending strength, MPa; $P$ is the axial load, N; $L$ is the length of the sample holder, 24 mm; $b$ is the width, 5 mm; and $h$ is the height, 8 mm.

A grinding ratio measurement device (DHM-2, Changchun, China) was applied to measure the grinding ratio. A schematic diagram of the grinding ratio test is illustrated in Figure 2. SiC grinding wheels with diameter of 100 mm and thickness of 20 mm were used as wear counterparts. The testing parameters involved were: linear velocity 15 m/s, swing frequency 30 min$^{-1}$, load 500 g, and grinding time ≥100 s. The grinding ratio $Ra$ was calculated by the formula:

$$Ra = \Delta Mg/\Delta Ms, \tag{2}$$

where $\Delta Mg$ is the weight loss of the SiC grinding wheel, and $\Delta Ms$ is the weight loss of sample.

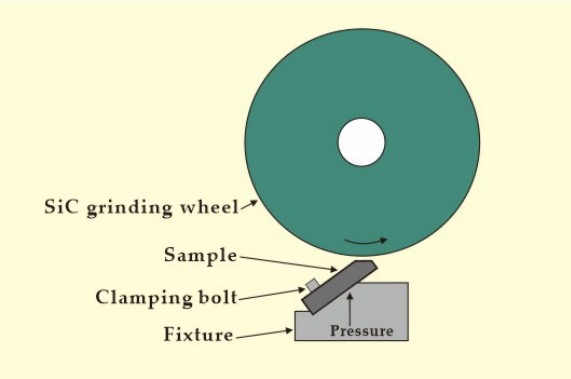

**Figure 2.** Schematic diagram of the grinding ratio test.

## 3. Results and Discussion

### 3.1. Composition and Microstructures

The XRD patterns of matrix samples with different deep cryogenic treatment times are shown in Figure 3. The elemental phase Fe, Cu, and intermetallic phases of $Co_3Fe_7$, FeCo, and $FeCu_4$ were observed in all samples. Note that the deep cryogenic treatment did not lead to an obvious change in the lattice parameters of phases, as determined on the basis of the XRD analysis.

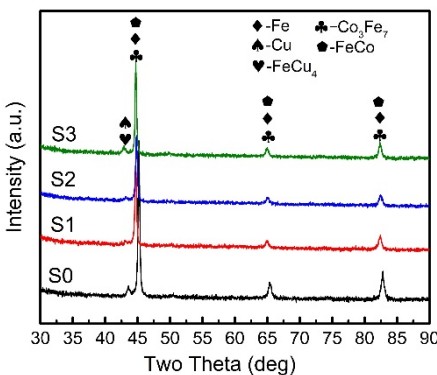

**Figure 3.** XRD patterns of matrix samples.

SEM micrographs for the bending fracture surface of matrix samples S0 and S3 are presented in Figure 4. The size of the grains was uniform and there was no abnormal grain growth. Examinations of the fracture surfaces indicated that the fracture mechanisms of matrix samples were trans-crystalline and inter-crystalline fracture, with inter-crystalline fracture being more common. It is important to note that the trans-crystalline fracture phenomenon of sample S3 (DCT for 3 h) was more obvious than in S0, which can be attributed to the development of stresses. The matrix material contained elemental phase Fe, Cu, and intermetallic phases of $Co_3Fe_7$, FeCo, and $FeCu_4$, which have different coefficients of thermal expansion (CTEs). During the deep cryogenic treatment, the high-CTE phase was subjected to compressive stresses and the low-CTE phase to tensile ones. These uneven structural changes that occurred because of normal contractions were opposed by transformation expansion at this temperature [26]. This phenomenon developed residual stresses in the matrix material, which led to an increase in dislocation density. When external stress was applied, the difficulty of grain dislocation motion increased, causing the grain to break and more easily spread inside the grain to produce trans-crystalline fracture.

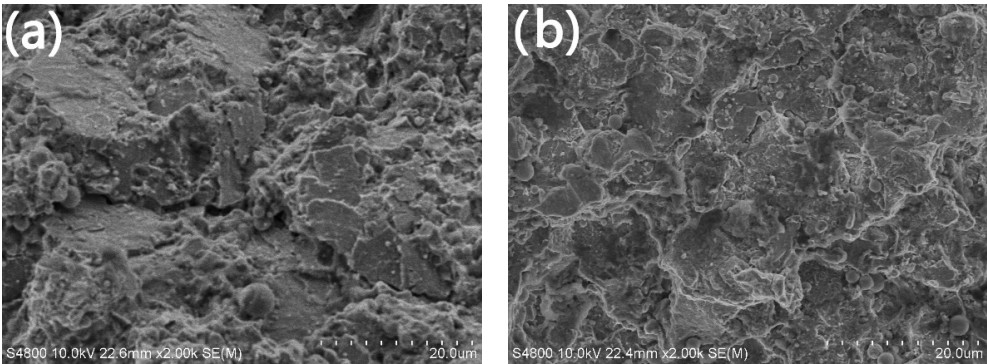

**Figure 4.** The fracture morphologies of samples (**a**) S0 and (**b**) S3.

SEM micrographs for the bending fracture surface of impregnated diamond samples SD0 and SD1 are given in Figure 5. The figure reveals that diamond grits were embedded in the metal matrix. As marked in Figure 5a,d, the average width of the interface crack between diamond grits and matrix decreased from 2.8 to 2.1 μm after deep cryogenic treatment for 1 h. The change of crack width can be attributed to the difference of thermal conductivity and thermal expansion coefficient of diamond and matrix materials. Table 3 shows the specific values of thermal conductivity and coefficient of thermal expansion [27]. During the deep cryogenic treatment process, the stress state changed significantly, and the matrix was subjected to tensile stress. At a certain moment, the tensile stress exceeded the micro-yield strength of the matrix, which led to the plastic deformation of the matrix materials. The smaller crack width indicates that the diamond grits were held more firmly by the matrix, which is conducive to the mechanical properties and wear resistance of impregnated diamond composites. The average crack widths of SD2 (DCT for 2 h) and SD3 (DCT for 3 h) were 2.6 and 2.5 μm, respectively. Cracks in the diamond/matrix interface have a negative impact on relative density, which is consistent with the test results in Table 2.

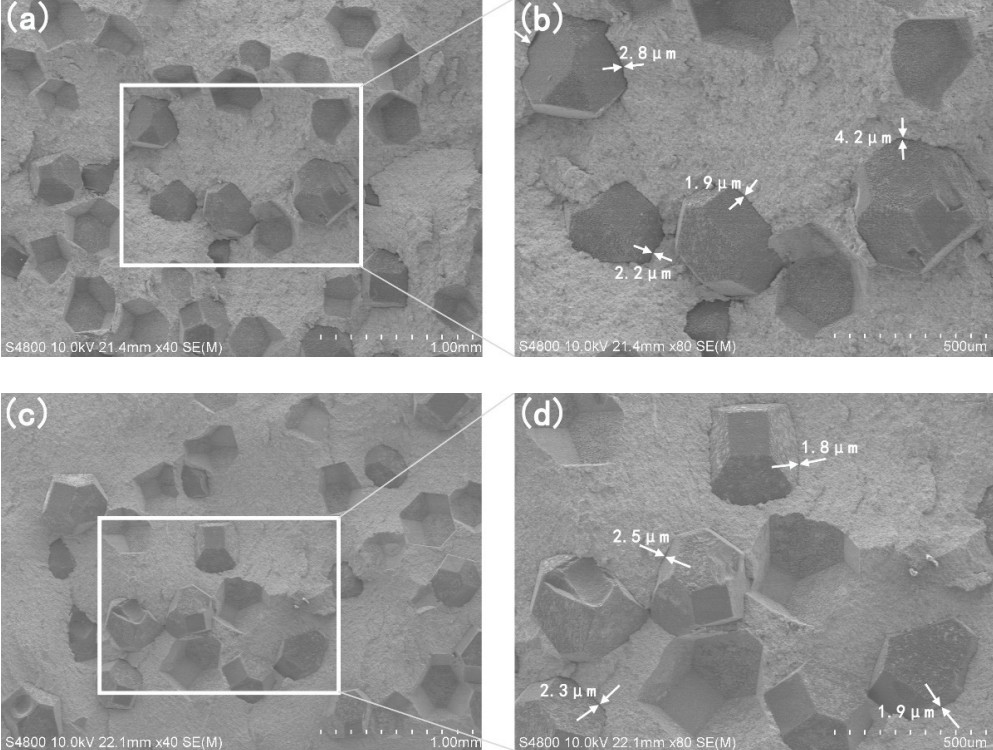

**Figure 5.** The fracture morphologies of impregnated samples: (**a**,**b**) SD0; (**c**,**d**) SD1. The crack widths were measured by image processing software at high magnification.

**Table 3.** The values of thermal conductivity and coefficient of thermal expansion.

| Composition | Thermal Conductivity (W/m·K) | Thermal Expansion Coefficient (K$^{-1}$) |
|---|---|---|
| Diamond | 2000 | $\approx 1 \times 10^{-6}$ |
| Matrix material | 125 | $\geq 13 \times 10^{-6}$ |

## 3.2. Mechanical Properties

The relative density of all tested samples is shown in Figure 6a. It can be seen that deep cryogenic treatment had little effect on the relative density of matrix samples. For impregnated samples, the relative density increased after deep cryogenic treatment due to the decrease of crack width between diamond and matrix.

As displayed in Figure 6b,c the hardness and bending strength of matrix samples increased slightly after deep cryogenic treatment. The results can be primarily related to the development of stresses. The development of stresses was due to the difference in thermal expansion coefficients between different phases. The development of stresses can increase the dislocation density, which leads to the improvement of the hardness and bending strength.

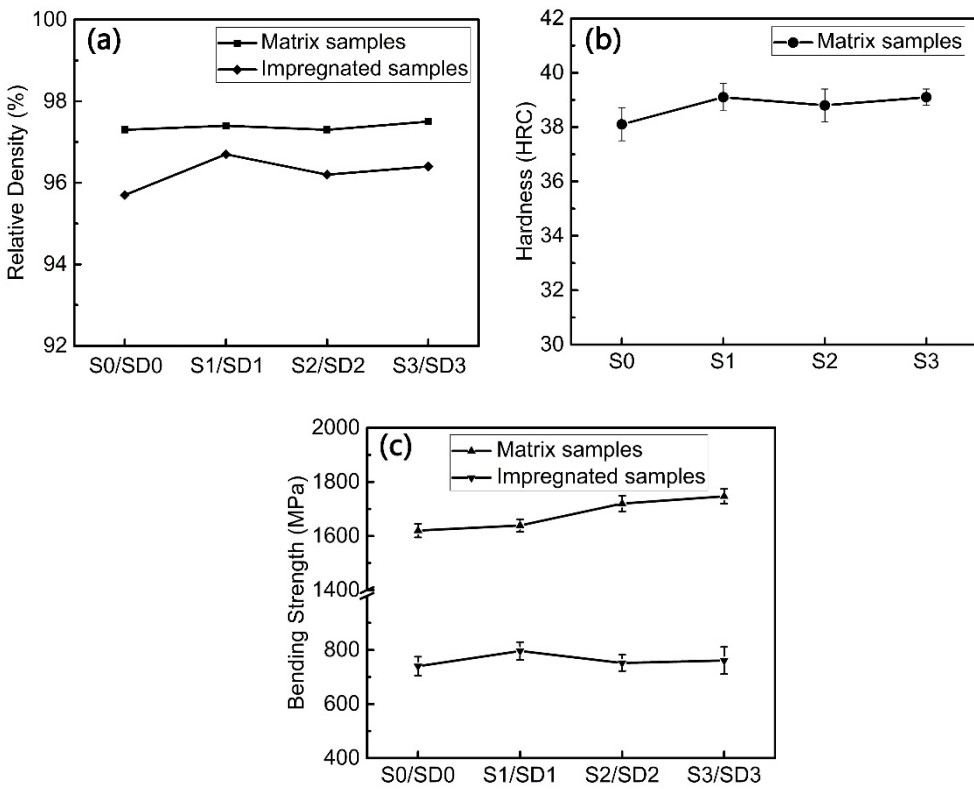

**Figure 6.** Mechanical properties of samples. (**a**) The relative density of all samples; (**b**) The HRC values of matrix samples; (**c**) The bending strength of all samples.

The bending strength of all samples is illustrated in Figure 6c. Sample SD1 showed the highest value and the values of SD2 and SD3 were similar to SD0. The interface structure in Figure 5 observed between diamond grits and matrix of all impregnated samples is concordant with the bending strength test results. The average crack width decreased from 2.8 to 2.1 μm after deep cryogenic treatment for 1 h. This structure feature means that the diamond grits were held by the matrix more firmly, which benefits the stress transfer, resulting in a high bending strength.

*3.3. Wear Resistance*

The grinding ratio is an important indicator to evaluate the wear resistance and tribological performance of diamond-impregnated composites [28,29]. The bar chart in Figure 7 summarizes the results for the grinding ratio of impregnated diamond samples with different deep cryogenic treatment times. The grinding ratio increased remarkably after deep cryogenic treatment, proving that deep cryogenic treatment improved the wear resistance of the impregnated diamond composites. With increasing deep cryogenic treatment time, the grinding ratio first increased and then decreased. The grinding ratio of sample SD1 showed a 32.9% increase in comparison with SD0.

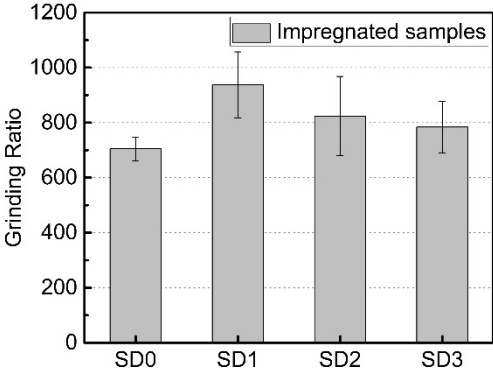

**Figure 7.** Results of the grinding ratio test of impregnated diamond samples.

The grinding ratio of impregnated samples depends largely on the bending strength. Some related studies analyzed the theoretical relationship between bending strength and wear resistance [5,30]. A lower bending strength means that the metal matrix supports the diamond grits weakly, leading to the pull-out of diamond grits from the matrix under complex stress conditions during working. Hence, the grinding ratio of SD1 subjected to 1 h of deep cryogenic treatment was higher than the reference one (SD0) without deep cryogenic treatment.

The grinding ratios of SD2 and SD3 were larger than that of SD0. However, the bending strengths of SD2 and SD3 were similar to that of SD0, indicating that there are other factors affecting the wear resistance. The mismatch in the coefficient of thermal expansion between diamond ($\approx 1 \times 10^{-6}$ K$^{-1}$) and matrix material ($\geq 13 \times 10^{-6}$ K$^{-1}$) needs to be given further consideration. During the deep cryogenic treatment process, the metal-matrix-impregnated diamond composites were subjected to a thermal shock, with a temperature drop of 215 °C, which increased the residual compressive stress on diamond grits. This change of the stress state increased the diamond retention capacity of the matrix. Furthermore, the matrix materials penetrated into the microcracks on the surface of diamond grits, which could also improve the diamond retention capacity of the matrix. Facing the action of complex alternating cutting force, diamond grits usually exhibit a tendency to rotate first and then pull out of the matrix easily. The improvement of the matrix's diamond retention capacity can reduce the rotating tendency, which is conducive to improving the wear resistance of diamond composites.

## 4. Conclusions

The effects of deep cryogenic treatment on the microstructure, mechanical properties, and wear resistance of Fe–Co–Cu-based diamond-impregnated composites were investigated. The experimental results indicated that the hardness and bending strength of matrix increased slightly after deep cryogenic treatment. The grinding ratio of the impregnated diamond composites increased remarkably by 32.9% as a result of deep cryogenic treatment. The strengthening mechanism was analyzed in detail. The conjoint effects of bending strength and diamond retention capability on the wear resistance of the diamond composites was revealed. The Fe–Co–Cu-based impregnated composites subjected to deep

cryogenic treatment for 1 h exhibited the optimal overall performance, thus providing guidance for further scientific research and actual applications.

**Author Contributions:** B.L. and S.L. conceived and designed the experiments; S.L. and W.D. performed the experiments; Z.H. and X.Z. analyzed the data; B.L. contributed analysis tools; S.L. wrote the paper.

**Funding:** This work was financially supported by the National Natural Science Foundation of China (41572357) and the Science and Technology project of Jilin Province Education Department (JJKH20180087KJ).

**Conflicts of Interest:** The authors declare no conflicts of interest.

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
