# Peer review of "Effect of Deep Cryogenic Treatment on Microstructure and Properties of Sintered Fe–Co–Cu-Based Diamond Composites"

_applsci, doi:10.3390/app9163353_

Round 1
Reviewer 1 Report
The effect of Deep Cryogenic Treatment on mechanical properties of sintered Fe-Co-Cu-based diamond composites is investigated in this study. The reported results and the discussions are reasonable. Thus, this study can be recommended for publication in Applied Sciences after the following points are fully addressed:
-The English level is not satisfactory in this manuscript. Multiple grammatical errors and typos can be seen in the manuscript which have decreased the quality of discussions. For instance, Line 42: “…, widely servers …”. Thus, a thorough proof reading of the manuscript preferably by professional English proof reading services is required.
Materials and methods:
-There is no explanation on why 840 °C was chosen as the sintering temperature ?
-How was the oxidation prevented at 840 °C ?
-Table 2 shows that hardness and bending strength of matrix increases as DCT duration increases. What happens when DCT continues beyond 3 hours ?
-What was the initial temperature of samples when DCT started ?
-The location of samples is missing in Figure 1.
Results and discussion:
-There is no explanation on why sample S3 showed trans-crystalline fracture as opposed to sample S0 ?
-The comparison between Figures 4a and b should be more detailed from morphological point of view.
-It is difficult to see the interface cracks in Figure 5. Higher magnification SEM images should be provided which clearly show the width of interface cracks.
-How were the thermal conductivity and thermal expansion coefficient measured ?
Author Response
AUG. 11, 2019
applsci-562272
Title: Effect of Deep Cryogenic Treatment on Microstructure and Properties of Sintered Fe-Co-Cu-based Diamond Composites
Dear editor and reviewers
Thank you for the comments. We have addressed all comments from the reviewer as follows:

Reviewer 2 Report
The reviewed article is very interesting. At present, the cryogenic method is a rapidly growing method of machining and treatment materials. An article is written clearly and the order of the chapters is correct. I have a questions:
High density sample was obtained after the hot isostatic pressing. But why did you use so low pressure for HIP (20 MPa)? Why did not you carry out grinding test for diamondless sample?Author Response
AUG. 11, 2019
applsci-562272
Title: Effect of Deep Cryogenic Treatment on Microstructure and Properties of Sintered Fe-Co-Cu-based Diamond Composites
Dear editor and reviewer
Thank you for the comments. We have addressed all comments from the reviewer as follows:
------------------------
Reviewers' comments:
Reviewer #2:
High density sample was obtained after the hot isostatic pressing. But why did you use so low pressure for HIP (20 MPa)?
Reply: Thanks for your suggestion. High density samples can be obtained under high pressure, but at the same time, high pressure will reduce the service life of graphite mould. In the diamond tool manufacturing industry, the hot-pressing sintering pressure of Fe based impregnated diamond composites is usually about 20 MPa. The sintering pressure of Fe based impregnated diamond composites also can be found in published literatures, such as reference [14].
Why did not you carry out grinding test for diamondless sample?Reply: In diamond tools industry, usually the grinding ratio of matrix samples is not measured. Because the grinding ratio of impregnated diamond samples reveals the wear resistance and performance of diamond tools comprehensively, the grinding ratio of matrix samples has less influence on diamond tool performance.
Professor Baochang Liu
College of Construction Engineering
Jilin University, Changchun 130012, China
E-mail: liubc@jlu.edu.cn